# The Impact of Reinitialization on Generalization in Convolutional Neural Networks

## Abstract

We study the impact of different reinitialization methods in several convolutional architectures for small-size image classification datasets. We analyze the potential gains of reinitialization and highlight limitations. We also study a new layerwise reinitialization algorithm that outperforms previous methods and suggest explanations of the observed improved generalization. First, we show that layerwise reinitialization increases the margin on the training examples without increasing the norm of the weights, hence leading to an improvement in margin-based generalization bounds for neural networks. Second, we demonstrate that it settles in flatter local minima of the loss surface. Third, it encourages learning general rules and discourages memorization by placing emphasis on the lower layers of the neural network.

## 1 Introduction

Deep neural networks demonstrate state-of-the-art performance on many classification tasks. While often highly overparameterized, modern deep architectures exhibit a remarkable ability to generalize beyond the training sample even when trained without an explicit form of regularization (Zhang et al., 2017). A large body of work has been devoted to offering insights into this "benign" overfitting phenomenon, including explanations based on the margin (Neyshabur et al., 2015; Bartlett et al., 2017; Neyshabur et al., 2017; Arora et al., 2018; Soudry et al., 2018), the curvature of the local minima (Keskar et al., 2017; Chaudhari et al., 2019; Neyshabur et al., 2020), and the speed of convergence (Hardt et al., 2016), among others.

Recently, however, a number of works suggest that generalization in convolutional neural networks (CNNs) could be improved further using reinitialization. Precisely, let $\mathbf{w} \in \mathbb{R}^d$ be a vector that contains all of the parameters in a neural network (e.g. filters in convolutional layers and weight matrices in fully-connected layers). Let $\mathbf{s} \in \{0, 1\}^d$ be a binary mask that is generated at random according to some probability mass function. Then, "reinitialization" is selecting a subset of parameters and reinitializing them during training:

$$\mathbf{w} \; \leftarrow \; (1 - \mathbf{s}) \odot \mathbf{w} \; + \; \mathbf{s} \odot \eta, \tag{1}$$

where $\odot$ is an element-wise multiplication and $\eta$ is a random initialization of the model parameters. For example, $\eta$ may correspond to the weights of He or Xavier initializations (He et al., 2015; Glorot & Bengio, 2010). In the following, we refer to the update in (1) as a "reinitialization round." Reinitialization methods differ in how the binary mask $\mathbf{s}$ is selected. Four prototypical approaches are:

- **Random subset**: A random subset of the parameters of a fixed size is chosen uniformly at random in each round. This includes, for example, the random weight level splitting (WELSR) method studied in (Taha et al., 2021), in which about 20% of the parameters are selected for reinitialization.

- **Weight magnitudes**: The smallest parameters in terms of their absolute magnitudes are reinitialized at each round. This can be interpreted as a generalization to the dense-sparse-dense (DSD) workflow of Han et al. (2017) in which reinitialization occurs only once.

- **Fixed subset**: A subset is chosen at random prior to training and is fixed afterwards. It corresponds to the weight level splitting (WELS) method of Taha et al. (2021).

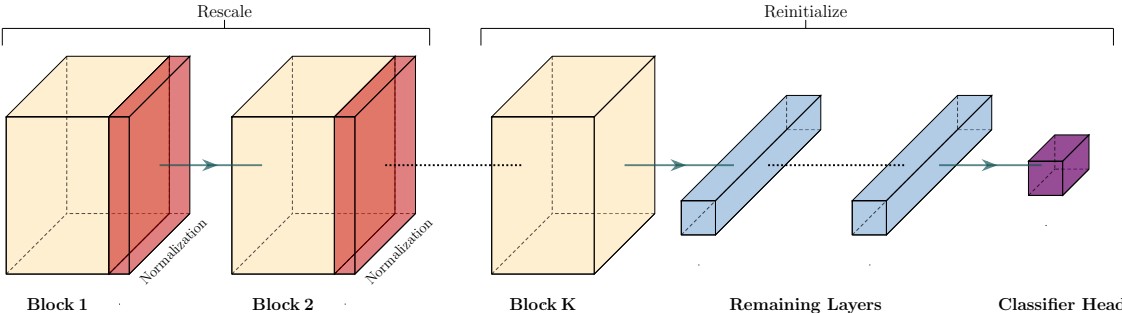

Figure 1: Given a deep neural network starting with $K$ convolutional blocks followed by other layers, LW proceeds sequentially from bottom to top (see Algorithm 1). When in block $k$ (e.g. $k = 2$ in the figure above), the weights of all early blocks $\{1, \ldots, k\}$ are rescaled while subsequent layers are reinitialized. In addition, a normalization layer is inserted following block $K$ shown in red. Importantly, the network may contain other normalization layers within each block, such as batch or layer normalization (Ioffe & Szegedy, 2015; Ba et al., 2016). Red layers correspond to the normalization layers inserted by LW, which are fixed (non-trainable) at each round.

- **Fully-connected layers**: Only the last fully-connected layers are reinitialized. This includes, for example, the method proposed in (Li et al., 2020). In (Zhao et al., 2018), only the classifier head is reinitialized.

We denote these four methods as WELSR, DSD, WELS, and FC, respectively. Moreover, we denote the baseline method of training once until convergence as BL.

In this paper, we also study a new reinitialization algorithm, which we denote as LW for its LayerWise approach. The new algorithm is motivated by the common observation that lower layers in the neural network tend to learn general rules while upper layers specialize (Yosinski et al., 2014; Arpit et al., 2017; Raghu et al., 2019; Maennel et al., 2020; Baldock et al., 2021). While all reinitialization methods improve generalization in CNNs, we demonstrate in Section 3 that LW often outperforms the other methods. It encourages learning general rules by placing more emphasis on training the early layers of the neural network. A more formal statement is presented in Section 3.

- **Layerwise**: A convolutional neural network is partitioned into $K$ blocks (see Figure 1 and Algorithm 1). At round $k$, the parameters at the lowest $k$ blocks are rescaled back to their original norm during initialization (see Algorithm 1) while the rest of the network is reinitialized. In addition, a new normalization layer is inserted/updated following block $k$. This is repeated for a total of $N \geq 1$ iterations for each block.

It is worth noting that FC is a special case of LW, in which $K = 1$ and $N > 1$. In addition, the concurrent work of Zhou et al. (2021) also corresponds to $K = 1$ where the upper $L$ layers are reinitialized at each round for some fixed $L > 1$. Besides the prominent role of reinitialization, LW includes normalization and rescaling, which we show in an ablation study in Appendix E to be important. In Appendix A, we discuss why LW can be interpreted as a stochastic gradient descent (SGD) procedure to a well-defined stochastic loss. Next, we illustrate the basic principles of these reinitialization methods on a minimal example with synthetic data.

## 1.1 Synthetic Data Example

**Setup.** Let $\mathbf{x} \in \mathbb{R}^{128}$ be the instance and $\mathbf{y}$ be its label, which is sampled uniformly at random from the set $\{0, 1, ..., 7\}$. For the instances, on the other hand, each of the first 3 coordinates of $\mathbf{x}$ is chosen from $\{-1, 1\}$ to encode the label $\mathbf{y}$ in binary form. For example, instances in class 0 would have their first three coordinates as $(-1, -1, -1)$, whereas instances in class 5 would have $(1, -1, 1)$. Consequently, the first three

---

**Algorithm 1** Pseudocode of LW

---

**Input:** (1) Neural network with identified sequence of $K \geq 1$ conv blocks; (2) Training dataset; (3) $N \geq 1$.

**Output:** Trained model parameters.

**Training:**

1: Initialize the neural network architecture;
2: For each layer $l$, compute $s_l = ||W_l||_2$, where $W_l$ are the weights of layer $l$.
3: **for** $k \in (1, 2, \ldots, K)$ **do**
4:     **for** $n \in (1, 2, \ldots, N)$ **do**
5:         **for** $j \in (1, 2, \ldots, k)$ **do** # *rescaling*
6:             **for** layer $\in$ Block $j$ **do**
7:                 $W_{\text{layer}} \leftarrow (s_{\text{layer}}/||W_{\text{layer}}||_2) \cdot W_{\text{layer}}$
8:             **end for**
9:         **end for**
10:         Pick a batch $X$ of training set uniformly at random;
11:         Compute $Z$: the output of Block $k$ of $X$;
12:         Compute $\mu, \sigma \in \mathbb{R}$: mean and standard deviation of $Z$;
13:         **if** $n = 1$ **then**
14:             Insert lambda layer $\lambda x : (x - \mu)/\sigma$ after block $k$;
15:         **else**
16:             Update lambda layer with new values of $\mu$ and $\sigma$;
17:         **end if**
18:         Reinitialize all layers above block $k$;
19:         Fine-tune the entire model until convergence;
20:     **end for**
21: **end for**

---

Table 1: Test accuracy [%] for the synthetic data experiment of Section 1.1 with different signal strengths $\alpha$ and different reinitialization methods. We observe that all initialization methods (with the exception of DSD) improve generalization in this example setting with LW performing best. In addition, reinitialization methods also tend to reduce the variance of the test accuracy.

| $\alpha$ | BL | WELSR | DSD | WELS | FC | LW |
|---|---|---|---|---|---|---|
| 0.5 | $20.3 \pm 0.6$ | $\mathbf{24.6 \pm 1.0}$ | $22.9 \pm 0.6$ | $23.1 \pm 1.4$ | $23.6 \pm 3.0$ | $\mathbf{25.2 \pm 0.8}$ |
| 1.0 | $50.7 \pm 5.4$ | $\mathbf{72.9 \pm 0.9}$ | $53.4 \pm 0.7$ | $66.1 \pm 2.1$ | $68.6 \pm 2.1$ | $\mathbf{72.3 \pm 3.6}$ |
| 2.0 | $94.6 \pm 2.0$ | $98.2 \pm 0.4$ | $90.3 \pm 1.4$ | $96.8 \pm 0.1$ | $99.0 \pm 0.2$ | $\mathbf{99.8 \pm 0.2}$ |

coordinates of an instance correspond to its "signal." The remaining 125 entries of **x** are randomly sampled i.i.d. from $\mathcal{N}(0, 1)$.

Although we focus in this work on convolutional neural networks, we use a multilayer perceptron (MLP) in this synthetic data experiment because the inputs are not images but generic feature vectors. The MLP contains two hidden layers of 32 neurons with ReLU activations (Nair & Hinton, 2010) followed by a classifier head with softmax activations. It optimizes the cross–entropy loss. We train on 256 examples using gradient descent with learning rate 0.05.

**Methods.** Treating every layer as a block, we have $K = 3$. If 200 training steps are used per round of reinitialization and $N = 3$, LW trains the model once for 200 steps after which the 2nd and 3rd layers are reinitialized (in addition to rescaling and normalization). This is carried out $N = 3$ times in the first layer before $k$ is incremented. The same process is repeated on each layer making a total of $200 \times N \times K = 1,800$ training steps overall. In WELS, WELSR, DSD, and FC, the model is trained for 200 steps before reinitialization is applied, and this is repeated $K \times N$ times for the same total of $1,800$ steps. The baseline method corresponds to training the model once without reinitialization for a total of $1,800$ training steps.

**Results.** When trained for 1,800 steps, the baseline (BL) achieves 100% training accuracy, but only around 51% test accuracy. The large gap between training and test accuracy for such a simple task is reminiscent of

the classical phenomenon of overfitting. Note that the number of training examples is 256, which is generally small for 128 features of equal variance. On the other hand, reinitialization improves accuracy as shown in Table 1 even though these reinitialization methods do *not* have access to any additional data and use the same optimizer and hyper-parameters as baseline training. The training accuracy is 100% in all cases. We also observe that reinitialization tends to reduce the variance of the accuracy (with respect to the seed).

In the above experiment, both the signal part (first three coordinates) and the noise part (remaining coordinates) have the same scale (standard deviation 1). We can make the classification problem easier or harder by multiplying the signal part by a signal strength $\alpha > 1$ or $\alpha < 1$, respectively. We present the average test accuracy in Table 1 for a selection of values of $\alpha$ with $N = 3$. Appendix B contains additional results when weight decay is added.

## 1.2 Contribution

In this work, we study a new layerwise reinitialization algorithm LW, which often outperforms other methods. We provide two explanations, supported by experiments, for why it improves generalization in convolutional neural networks. First, we show that LW improves the margin on the training examples without increasing the norm of the weights, hence leading to an improvement in known margin-based generalization bounds in neural networks. Second, we show that LW settles in flatter local minima of the loss surface.

Furthermore, we provide a comprehensive study comparing previous reinitialization methods: First, we evaluate different methods within the same context. For example, the comparison in (Taha et al., 2021) uses only a single reinitialization round of the dense-sparse-dense approach (DSD) when DSD can be extended to multiple rounds. Also, (Zhao et al., 2018) uses an ensemble of classifiers when reinitializing the fully-connected layers, which could (at least partially) explain the improvement in performance. By contrast, we follow a coherent training protocol for all methods. Second, we use our empirical evaluation to analyze the effect of the experiment's design, such as augmentation, dropout and momentum. The goal is to determine if the effect of reinitialization could be achieved by tuning such settings. Third, we employ decision tree classifiers to identify when each reinitialization method is likely to outperform others. In summary, we:

1. Study a new reinitialization method, denoted LW, which is motivated by common observations of generalization and memorization effects across the neural network's layers. We show that it outperforms other methods with a statistically significant evidence at the 95% level.

2. Suggest two explanations, supported by experiments, for why LW is more successful at improving generalization in CNNs compared to other methods.

3. Present a comprehensive evaluation study of reinitialization methods covering more than 1,000 experiments for four convolutional architectures: (1) simplified CNN, (2) VGG16 (Simonyan & Zisserman, 2015), (3) MobileNet (Howard et al., 2017) and (4) ResNet50 (He et al., 2016a). We conduct the evaluation over 6 benchmark image classification datasets of small size (up to 12,000 training examples per dataset). We do not observe consistent gains of reinitialization with large datasets, so we omit those from the comparison and focus on the small-data regime.

## 2 Related Work

**Reinitialization.** As stated earlier, a number of works suggest that reinitializing a subset of the neural network parameters during training can improve generalization. This includes, the dense-sparse-dense (DSD) training workflow proposed by Han et al. (2017), in which reinitialization occurs only once during training. However, as the authors argue, the improvement in accuracy in DSD could be attributed to the effect of introducing sparsity, not reinitialization. Another example is "Knowledge Evolution", including weight level splitting (WELS) and its randomized version (WELSR) (Taha et al., 2021). It was noted that WELS outperformed WELSR, which agrees with our observations. Finally, some recent works propose to reinitialize the fully-connected layers only (Li et al., 2020; Zhao et al., 2018). In particular, reinitializing the last layer several times and combining the models into an ensemble can improve performance (Zhao et al., 2018).

However, the improvement in accuracy could (at least partially) be attributed to the ensemble of predictors, not to reinitialization *per se*. For fair comparison, we extend DSD to multiple rounds of reinitialization and do not use an ensemble of predictors.

**Generalization Bounds.** Several generalization bounds for neural networks have been proposed in the literature. Of those, a prototypical approach is to bound the generalization gap by a particular measure of the *size of weights* normalized by the *margin* on the training set. Examples of measures of the size of weights include the product of the $\ell_1$ norms (Bartlett, 1998) and the product of the Frobenius norms of layers (Neyshabur et al., 2015), among others (Bartlett et al., 2017; Neyshabur et al., 2017; Arora et al., 2018). While such generalization bounds are often loose, they were found to be useful for ranking models (Neyshabur et al., 2017). The fact that rich hypothesis spaces could still generalize if they yield a large margin over the training set was used previously to explain the performance of boosting (Schapire et al., 1997). In Section 4, we show that LW boosts the margin on the training examples without increasing the size of the weights.

**Flatness of the Local Minimum.** Another important line of work examines the connection between generalization and the curvature of the loss at the local minimum (Keskar et al., 2017; Neyshabur et al., 2017; Foret et al., 2021). Deep neural networks are known to converge to local minima with sparse eigenvalues (>94% zeros) in their Hessian (Chaudhari et al., 2019). Informally, a flat local minimum is robust to data perturbation, and this robustness can, in turn, be connected to regularization (Bishop, 1995). In fact, some of the benefits of transfer learning were attributed to the flatness of the local minima (Neyshabur et al., 2020). For a precise treatment, one may use the PAC-Bayes framework to derive a generalization bound that comprises of two terms: (1) sharpness of the local minimum, and (2) the weight norm over noise ratio (Neyshabur et al., 2017). Similar terms also surface in the notion of "local entropy" (Chaudhari et al., 2019). We show in Section 4 that LW improves both terms.

**Generalization vs. Memorization.** Several works point out that early layers in a neural network tend to learn general-purpose representations whereas later layers specialize, e.g. (Raghu et al., 2019; Arpit et al., 2017; Yosinski et al., 2014; Maennel et al., 2020). This can be observed, for instance, using probes, in which classifiers are trained on the layer embeddings. As demonstrated in (Cohen et al., 2018) and (Baldock et al., 2021), deep neural networks learn to separate classes at the early layers with real labels (generalization) but they only separate classes at later layers when the labels are random (memorization). One explanation for why LW improves generalization is that it encourages learning general rules at early layers and discourages memorization at later layers.

## 3 Analysis

### 3.1 Empirical Study

We begin by evaluating the performance of the five reinitialization methods discussed in Section 1 for four convolutional architectures on 6 small-size benchmark image classification datasets (see Table 4 for details). Appendix F summarizes related experiments on CIFAR10 and CIFAR100. All images are resized to 224×224. The architectures are (1) simplified CNN, (2) VGG16 (Simonyan & Zisserman, 2015), (3) MobileNet (Howard et al., 2017) and (4) ResNet50 (He et al., 2016a). We denote these by `scnn`, `vgg16`, `mobilenet`, and `resnet50`, respectively. We use He-initialization (He et al., 2015) unless stated otherwise. Details about each architecture and the hyper-parameters are provided in Appendix C.

To recall, every reinitialization method trains the same model on the same dataset for several rounds. After each round, a binary mask of the model parameters is selected according to the reinitialization criteria and the update in Eq. (1) is applied for some random initialization $\eta$. After that, the model is fine-tuned on the same data. Blocks in LW correspond to the standard blocks of the architecture (e.g. a block in Figure 1 would correspond to either an identity or a convolutional block in ResNet50). Also, 10% of the training split is reserved for validation, which is used for early stopping in all methods.

To evaluate the relative performance of the reinitialization methods, we perform a set of experiments in which we *fix* the hyperparameters for all architectures and datasets to the same values. The hyperparameters were

Table 2: Test accuracy results [%] for the five reinitialization methods on the six benchmark datasets: Oxford-IIIT (Parkhi et al., 2012a), Stanford Dogs (Deng et al., 2009), Cars (Krause et al., 2013b), Caltech-101 (Fei-Fei et al., 2004b), Cassava (Mwebaze et al., 2019a), and Caltech-UCSD Birds 200 (Welinder et al., 2010a). Values in **bold**/underlined are the **best**/second-best results. The symbols B,R,D,W,F,L are for baseline, WELSR, DSD, WELS, FC, and LW, respectively. In LW, $N = 1$. Every reinitialization uses the same number of rounds $K$ (cf. Appendix C) and they are trained for the same number of epochs (including the baseline). In WELS, WELSR, and DSD, we reinitialize 20% of the parameters, following Taha et al. (2021).

| | | B | R | D | W | F | L | | B | R | D | W | F | L |
|---|---|---|---|---|---|---|---|---|---|---|---|---|---|---|
| OXFORD-IIIT | scnn | $13.7^{\pm.9}$ | $12.0^{\pm.6}$ | $13.4^{\pm.2}$ | $\underline{14.0^{\pm.6}}$ | $\underline{14.0^{\pm.4}}$ | $\mathbf{15.0^{\pm.4}}$ | vgg16 | $16.2^{\pm.1}$ | $16.7^{\pm.9}$ | $16.9^{\pm.9}$ | $16.4^{\pm.5}$ | $\mathbf{29.7^{\pm.8}}$ | $\underline{25.6^{\pm.9}}$ |
| | mobile | $13.4^{\pm.2}$ | $13.8^{\pm.6}$ | $\underline{17.2^{\pm.1}}$ | $15.9^{\pm2.}$ | $14.8^{\pm3.}$ | $\mathbf{30.1^{\pm.8}}$ | resnet | $24.0^{\pm.1}$ | $24.2^{\pm.6}$ | $22.9^{\pm.1}$ | $\underline{24.9^{\pm.2}}$ | $23.7^{\pm1.}$ | $\mathbf{28.5^{\pm.1}}$ |
| DOGS | scnn | $\underline{5.8^{\pm.3}}$ | $5.2^{\pm.5}$ | $5.2^{\pm.1}$ | $5.1^{\pm.5}$ | $5.1^{\pm.1}$ | $\mathbf{6.1^{\pm.2}}$ | vgg16 | $11.4^{\pm.2}$ | $11.8^{\pm.1}$ | $10.7^{\pm.4}$ | $10.3^{\pm.1}$ | $\mathbf{24.2^{\pm.6}}$ | $\underline{19.1^{\pm.9}}$ |
| | mobile | $9.7^{\pm.3}$ | $8.6^{\pm1.}$ | $9.0^{\pm1.}$ | $\underline{11.6^{\pm.4}}$ | $9.9^{\pm.5}$ | $\mathbf{19.2^{\pm1.}}$ | resnet | $14.0^{\pm.1}$ | $17.5^{\pm1.}$ | $\underline{19.2^{\pm2.}}$ | $16.0^{\pm1.}$ | $13.9^{\pm0.6}$ | $\mathbf{22.0^{\pm.7}}$ |
| CARS196 | scnn | $3.4^{\pm.1}$ | $3.4^{\pm.0}$ | $3.4^{\pm.4}$ | $\underline{3.5^{\pm.3}}$ | $3.2^{\pm.1}$ | $\mathbf{3.8^{\pm.1}}$ | vgg16 | $6.5^{\pm1.}$ | $5.5^{\pm.1}$ | $7.7^{\pm.4}$ | $7.0^{\pm.1}$ | $\mathbf{20.9^{\pm.1}}$ | $\underline{9.9^{\pm.5}}$ |
| | mobile | $6.1^{\pm2.}$ | $\underline{8.9^{\pm1.}}$ | $5.3^{\pm2.}$ | $7.4^{\pm.5}$ | $6.8^{\pm1.}$ | $\mathbf{22.2^{\pm.1}}$ | resnet | $10.1^{\pm2.}$ | $12.8^{\pm.6}$ | $\underline{13.4^{\pm6.}}$ | $\mathbf{13.7^{\pm1.}}$ | $10.4^{\pm.1}$ | $12.6^{\pm.3}$ |
| CALTECH101 | scnn | $\mathbf{51.1^{\pm.4}}$ | $49.9^{\pm.1}$ | $49.8^{\pm.8}$ | $48.4^{\pm.9}$ | $\underline{50.8^{\pm.7}}$ | $50.7^{\pm.1}$ | vgg16 | $54.1^{\pm.3}$ | $55.4^{\pm.7}$ | $54.2^{\pm.5}$ | $55.9^{\pm.3}$ | $\mathbf{69.1^{\pm.1}}$ | $\underline{57.3^{\pm1}}$ |
| | mobile | $36.7^{\pm.9}$ | $43.0^{\pm3.}$ | $\underline{44.9^{\pm.4}}$ | $40.9^{\pm3.}$ | $42.3^{\pm1.}$ | $\mathbf{47.0^{\pm.7}}$ | resnet | $50.4^{\pm.8}$ | $\underline{54.6^{\pm.8}}$ | $\mathbf{55.0^{\pm.4}}$ | $52.9^{\pm2.}$ | $50.7^{\pm1.}$ | $53.3^{\pm3.}$ |
| CASSAVA | scnn | $59.4^{\pm.4}$ | $58.6^{\pm.1}$ | $58.6^{\pm.7}$ | $59.8^{\pm1.}$ | $\underline{61.6^{\pm.1}}$ | $\mathbf{63.9^{\pm.3}}$ | vgg16 | $58.4^{\pm.2}$ | $\underline{59.5^{\pm1.}}$ | $58.4^{\pm2.}$ | $\mathbf{62.1^{\pm2.}}$ | $58.5^{\pm.0}$ | $59.3^{\pm1.}$ |
| | mobile | $52.6^{\pm.7}$ | $57.9^{\pm2.}$ | $\underline{63.6^{\pm.3}}$ | $62.2^{\pm.4}$ | $55.4^{\pm1.}$ | $\mathbf{70.0^{\pm2.}}$ | resnet | $61.9^{\pm4.}$ | $57.3^{\pm1.}$ | $\underline{62.8^{\pm2.}}$ | $58.1^{\pm.4}$ | $56.7^{\pm2.}$ | $\mathbf{63.9^{\pm2.}}$ |
| BIRDS2010 | scnn | $2.0^{\pm.1}$ | $\mathbf{2.6^{\pm.3}}$ | $2.3^{\pm.1}$ | $2.2^{\pm.6}$ | $\underline{2.4^{\pm.3}}$ | $2.4^{\pm.1}$ | vgg16 | $3.8^{\pm.4}$ | $4.3^{\pm.6}$ | $4.1^{\pm.4}$ | $3.4^{\pm.2}$ | $\mathbf{8.5^{\pm.6}}$ | $\underline{5.1^{\pm2.}}$ |
| | mobile | $4.5^{\pm1.}$ | $3.9^{\pm.3}$ | $5.2^{\pm.4}$ | $\underline{5.9^{\pm.6}}$ | $5.3^{\pm1.}$ | $\mathbf{8.1^{\pm.5}}$ | resnet | $6.9^{\pm.4}$ | $\underline{8.7^{\pm.3}}$ | $10.0^{\pm.1}$ | $10.0^{\pm.5}$ | $6.5^{\pm.1}$ | $\mathbf{10.0^{\pm.7}}$ |
| **+ Augmentation** | | | | | | | | | | | | | | |
| OXFORD-IIIT | scnn | $15.1^{\pm.5}$ | $14.6^{\pm.6}$ | $14.3^{\pm.5}$ | $14.7^{\pm.4}$ | $\underline{16.0^{\pm3.}}$ | $\mathbf{16.6^{\pm3.}}$ | vgg16 | $22.0^{\pm3.}$ | $20.0^{\pm2.}$ | $20.9^{\pm3.}$ | $20.7^{\pm3.}$ | $\mathbf{39.1^{\pm3.}}$ | $\underline{29.8^{\pm4.}}$ |
| | mobile | $27.7^{\pm1.}$ | $\underline{24.1^{\pm1.}}$ | $23.2^{\pm2.}$ | $\underline{24.3^{\pm1.}}$ | $23.5^{\pm1.}$ | $\mathbf{41.7^{\pm1.}}$ | resnet | $29.7^{\pm.3}$ | $33.3^{\pm.3}$ | $39.1^{\pm.2}$ | $31.9^{\pm.4}$ | $\mathbf{36.6^{\pm.3}}$ | $\underline{34.4^{\pm.3}}$ |
| DOGS | scnn | $7.4^{\pm.3}$ | $\underline{8.2^{\pm.2}}$ | $7.7^{\pm.3}$ | $8.0^{\pm.1}$ | $8.1^{\pm.2}$ | $\mathbf{8.6^{\pm.3}}$ | vgg16 | $16.0^{\pm4.}$ | $16.9^{\pm3.}$ | $15.5^{\pm2.}$ | $16.7^{\pm4.}$ | $\mathbf{37.3^{\pm4.}}$ | $\underline{31.2^{\pm9.}}$ |
| | mobile | $19.0^{\pm.8}$ | $19.5^{\pm.5}$ | $\underline{27.1^{\pm.8}}$ | $18.8^{\pm.7}$ | $20.7^{\pm.3}$ | $\mathbf{35.8^{\pm.8}}$ | resnet | $26.4^{\pm1.}$ | $30.7^{\pm1.}$ | $28.4^{\pm1.}$ | $\underline{35.2^{\pm2.}}$ | $33.3^{\pm1.}$ | $\mathbf{36.9^{\pm1.}}$ |
| CARS | scnn | $5.6^{\pm.2}$ | $\underline{6.0^{\pm.5}}$ | $5.3^{\pm.1}$ | $5.5^{\pm.2}$ | $\mathbf{7.3^{\pm.3}}$ | $\underline{5.8^{\pm.2}}$ | vgg16 | $11.7^{\pm.4}$ | $10.7^{\pm.4}$ | $11.2^{\pm.4}$ | $11.4^{\pm.1}$ | $\mathbf{43.6^{\pm.2}}$ | $\underline{22.4^{\pm.4}}$ |
| | mobile | $6.9^{\pm2.}$ | $\underline{16.2^{\pm1.}}$ | $9.1^{\pm1.}$ | $11.9^{\pm1.}$ | $13.8^{\pm1.}$ | $\mathbf{44.0^{\pm1.}}$ | resnet | $21.8^{\pm1.}$ | $42.5^{\pm2.}$ | $33.2^{\pm1.}$ | $\underline{43.1^{\pm2.}}$ | $36.7^{\pm2.}$ | $\mathbf{43.6^{\pm1.}}$ |
| CALTECH101 | scnn | $50.1^{\pm.5}$ | $52.2^{\pm.5}$ | $52.2^{\pm.5}$ | $51.5^{\pm.5}$ | $\mathbf{54.4^{\pm.5}}$ | $\underline{52.8^{\pm.4}}$ | vgg16 | $55.9^{\pm1.}$ | $55.8^{\pm3.}$ | $57.1^{\pm3.}$ | $56.3^{\pm3.}$ | $\mathbf{67.1^{\pm2.}}$ | $\underline{59.1^{\pm3.}}$ |
| | mobile | $41.0^{\pm2.}$ | $41.5^{\pm2.}$ | $\underline{47.4^{\pm3.}}$ | $41.8^{\pm3.}$ | $\mathbf{48.0^{\pm1.}}$ | $46.3^{\pm1.}$ | resnet | $50.2^{\pm2.}$ | $51.9^{\pm1.}$ | $\underline{53.1^{\pm.7}}$ | $\mathbf{57.5^{\pm1.}}$ | $52.1^{\pm2.}$ | $50.5^{\pm.9}$ |
| CASSAVA | scnn | $58.9^{\pm.5}$ | $60.6^{\pm.1}$ | $59.4^{\pm.2}$ | $62.2^{\pm.5}$ | $\underline{67.0^{\pm.4}}$ | $\mathbf{69.3^{\pm.7}}$ | vgg16 | $70.3^{\pm2.}$ | $70.0^{\pm1.}$ | $70.0^{\pm1.}$ | $\underline{71.2^{\pm1.}}$ | $\mathbf{71.5^{\pm1.}}$ | $68.3^{\pm4.}$ |
| | mobile | $62.3^{\pm1.}$ | $72.8^{\pm.7}$ | $\mathbf{80.1^{\pm.8}}$ | $76.1^{\pm1.7}$ | $\underline{77.3^{\pm.5}}$ | $\mathbf{81.1^{\pm.4}}$ | resnet | $46.5^{\pm2.}$ | $79.2^{\pm2.}$ | $\mathbf{82.9^{\pm2.}}$ | $\underline{82.6^{\pm1.}}$ | $77.6^{\pm2.}$ | $73.9^{\pm2.}$ |
| BIRDS2010 | scnn | $3.8^{\pm.5}$ | $3.7^{\pm.2}$ | $\underline{4.0^{\pm.3}}$ | $3.2^{\pm.2}$ | $\mathbf{4.2^{\pm.1}}$ | $3.8^{\pm.1}$ | vgg16 | $5.5^{\pm1.}$ | $6.5^{\pm1.}$ | $5.6^{\pm.7}$ | $5.8^{\pm.9}$ | $\mathbf{13.2^{\pm1.}}$ | $\underline{9.8^{\pm2.}}$ |
| | mobile | $6.6^{\pm.5}$ | $9.0^{\pm.7}$ | $8.1^{\pm.7}$ | $6.2^{\pm.4}$ | $7.4^{\pm.7}$ | $\mathbf{9.8^{\pm.7}}$ | resnet | $8.5^{\pm.4}$ | $\underline{12.5^{\pm.4}}$ | $11.2^{\pm.4}$ | $\mathbf{13.0^{\pm.6}}$ | $11.9^{\pm.4}$ | $10.6^{\pm.5}$ |
| **+ Augmentation + Dropout** | | | | | | | | | | | | | | |
| OXFORD-IIIT | scnn | $15.8^{\pm.5}$ | $13.6^{\pm.5}$ | $14.3^{\pm.3}$ | $15.1^{\pm.5}$ | $\mathbf{19.3^{\pm.5}}$ | $\underline{17.1^{\pm.5}}$ | vgg16 | $25.3^{\pm3.}$ | $26.7^{\pm1.}$ | $26.6^{\pm2.}$ | $26.4^{\pm3.}$ | $\mathbf{43.4^{\pm3.}}$ | $\underline{34.4^{\pm5.}}$ |
| | mobile | $28.4^{\pm1.}$ | $29.1^{\pm1.}$ | $27.8^{\pm1.}$ | $27.9^{\pm1.}$ | $22.0^{\pm1.}$ | $\mathbf{41.0^{\pm1.}}$ | resnet | $33.1^{\pm.3}$ | $35.2^{\pm.2}$ | $\mathbf{41.2^{\pm.3}}$ | $\underline{37.4^{\pm.3}}$ | $32.6^{\pm.3}$ | $35.6^{\pm.3}$ |
| DOGS | scnn | $8.4^{\pm.3}$ | $8.9^{\pm.3}$ | $7.6^{\pm.3}$ | $8.0^{\pm.3}$ | $\mathbf{9.6^{\pm.3}}$ | $\underline{9.1^{\pm.3}}$ | vgg16 | $17.7^{\pm4.}$ | $19.7^{\pm4.}$ | $19.5^{\pm3.}$ | $18.9^{\pm3.}$ | $\underline{34.9^{\pm7.}}$ | $\mathbf{35.9^{\pm9.}}$ |
| | mobile | $17.8^{\pm.8}$ | $23.5^{\pm.5}$ | $\underline{27.3^{\pm.9}}$ | $22.4^{\pm.8}$ | $20.5^{\pm.5}$ | $\mathbf{35.0^{\pm.4}}$ | resnet | $30.7^{\pm1.}$ | $33.3^{\pm1.}$ | $33.1^{\pm.9}$ | $33.8^{\pm.9}$ | $\underline{34.5^{\pm.9}}$ | $\mathbf{40.1^{\pm.9}}$ |
| CARS | scnn | $6.3^{\pm.1}$ | $6.2^{\pm.2}$ | $5.9^{\pm.1}$ | $\underline{6.8^{\pm.2}}$ | $\mathbf{7.4^{\pm.1}}$ | $6.4^{\pm.2}$ | vgg16 | $14.3^{\pm.7}$ | $18.9^{\pm.3}$ | $12.6^{\pm.3}$ | $16.6^{\pm.4}$ | $\mathbf{45.2^{\pm.1}}$ | $\underline{34.2^{\pm.5}}$ |
| | mobile | $9.5^{\pm1.}$ | $16.1^{\pm1.}$ | $\underline{30.8^{\pm1.}}$ | $24.1^{\pm2.}$ | $16.4^{\pm1.}$ | $\mathbf{44.5^{\pm1.}}$ | resnet | $19.8^{\pm2.}$ | $45.7^{\pm1.}$ | $43.0^{\pm1.}$ | $\mathbf{48.1^{\pm3.}}$ | $45.0^{\pm3.}$ | $\underline{47.5^{\pm2.}}$ |
| CALTECH101 | scnn | $\underline{51.4^{\pm.4}}$ | $\mathbf{53.5^{\pm.3}}$ | $51.1^{\pm.3}$ | $52.6^{\pm.1}$ | $52.4^{\pm.4}$ | $\mathbf{53.6^{\pm.3}}$ | vgg16 | $59.6^{\pm2.}$ | $61.7^{\pm3.}$ | $60.8^{\pm1.}$ | $61.0^{\pm3.}$ | $\mathbf{68.1^{\pm4.}}$ | $\underline{62.7^{\pm7.}}$ |
| | mobile | $47.1^{\pm2.}$ | $43.0^{\pm1.}$ | $47.5^{\pm1.}$ | $45.6^{\pm1.}$ | $\mathbf{51.3^{\pm1.}}$ | $\underline{49.4^{\pm2.}}$ | resnet | $50.6^{\pm2.}$ | $54.1^{\pm3.}$ | $\underline{54.6^{\pm1.}}$ | $\mathbf{55.7^{\pm1.}}$ | $51.4^{\pm2.}$ | $50.3^{\pm1.}$ |
| CASSAVA | scnn | $61.8^{\pm.5}$ | $62.9^{\pm.3}$ | $60.7^{\pm.2}$ | $61.9^{\pm1.}$ | $\underline{68.4^{\pm.1}}$ | $\mathbf{68.6^{\pm.1}}$ | vgg16 | $70.1^{\pm2.}$ | $71.4^{\pm2.}$ | $\mathbf{73.7^{\pm3.}}$ | $71.0^{\pm2.}$ | $71.1^{\pm2.}$ | $\underline{71.9^{\pm5.}}$ |
| | mobile | $68.5^{\pm1.}$ | $70.0^{\pm1.}$ | $\underline{78.6^{\pm1.}}$ | $74.3^{\pm1.}$ | $73.7^{\pm1.}$ | $\mathbf{80.5^{\pm3.}}$ | resnet | $74.2^{\pm2.}$ | $77.6^{\pm2.}$ | $\mathbf{81.5^{\pm1.}}$ | $\underline{80.9^{\pm1.}}$ | $73.5^{\pm2.}$ | $78.6^{\pm1.}$ |
| BIRDS2010 | scnn | $\underline{4.0^{\pm.2}}$ | $\underline{4.0^{\pm.1}}$ | $3.5^{\pm.2}$ | $3.7^{\pm.2}$ | $3.3^{\pm.1}$ | $\mathbf{4.4^{\pm.2}}$ | vgg16 | $6.5^{\pm3}$ | $8.1^{\pm1.}$ | $8.5^{\pm2.}$ | $8.1^{\pm1.}$ | $\mathbf{18.6^{\pm1.}}$ | $\underline{12.6^{\pm1.}}$ |
| | mobile | $2.5^{\pm.7}$ | $\mathbf{9.6^{\pm.7}}$ | $5.1^{\pm.6}$ | $6.3^{\pm.7}$ | $8.4^{\pm.2}$ | $\underline{9.0^{\pm.9}}$ | resnet | $11.5^{\pm.4}$ | $\underline{14.8^{\pm.6}}$ | $13.3^{\pm.7}$ | $\mathbf{15.2^{\pm1.}}$ | $13.5^{\pm.7}$ | $13.3^{\pm.3}$ |

Table 3: Statistical significance: a star ($\star$) implies that the column method outperforms the row with statistically significant evidence at the 95% level, computed using the exact binomial test. A circle ($\circ$) implies that statistical significance holds even after applying Holm's step-down correction for multiple hypothesis tests (Demšar, 2006). Only LW performs better than the baseline across all architectures. For `resnet50`, reinitialization methods except FC perform better than the baseline with no clear winner among them.

| | scnn | | | | | | vgg16 | | | | | | mobilenet | | | | | | resnet50 | | | | | |
|---|---|---|---|---|---|---|---|---|---|---|---|---|---|---|---|---|---|---|---|---|---|---|---|---|
| | B | R | D | W | F | L | B | R | D | W | F | L | B | R | D | W | F | L | B | R | D | W | F | L |
| B | | | | | | ○ | | | | ★ | ○ | ○ | | | | | | ○ | | ★ | ○ | ○ | | ★ |
| R | | | | | | ★ | | | | | ★ | ★ | | | | | | ★ | | | | ★ | | |
| D | | | | | | ★ | | | | | ★ | ★ | | | | | | ★ | | | | | | |
| W | | | | | ★ | ★ | | | | | ○ | ★ | | | | | | ○ | | | | | | |
| F | | | | | | | | | | | | | | | | | | ★ | | | | | | |
| L | | | | | | | | | | | | ★ | | | | | | | | | | | | |

Table 4: Overview of the 6 benchmark datasets.

| Name | \|Training\| | \|Test\| | # Classes |
|---|---|---|---|
| OXFORD-IIIT (Parkhi et al., 2012b) | 3,680 | 3,669 | 37 |
| DOGS (Khosla et al., 2011) | 12,000 | 8,580 | 120 |
| CARS196 (Krause et al., 2013a) | 8,144 | 8,041 | 196 |
| CALTECH101 (Fei-Fei et al., 2004a) | 3,060 | 6,084 | 101 |
| CASSAVA (Mwebaze et al., 2019b) | 5,656 | 1,885 | 4 |
| BIRDS2010 (Welinder et al., 2010b) | 3,000 | 3,033 | 200 |

chosen to work reasonably well across all combinations; in particular they enable reaching 100% training accuracy in all cases. We use SGD with an initial learning rate of 0.003 and momentum 0.9. The learning rate is decreased by a factor of 2 whenever the validation error does not improve for 20 epochs. The batch size is 256 and a maximum of 100k minibatch steps are used. We run all experiments, as explicitly stated, without data augmentation or with mild augmentation consisting of horizontal flipping and random cropping (in which the size is increased to $248 \times 248$ before a crop of size $224 \times 224$ is selected). Such fixed hyperparameters are suboptimal for some combinations of architectures and datasets and therefore the resulting numbers can be worse than state-of-the-art results. However, they enable reaching 100% training accuracy in all combinations of models and datasets. For example, increasing the learning rate to 0.01 would prevent ResNet50 from progressing its training error beyond that of random guessing on CASSAVA.

Table 2 provides the detailed results of the five reinitialization methods across the benchmark datasets in three settings: (1) no augmentation or dropout is used, (2) with augmentation, and (3) with both augmentation and dropout rate 0.25. We perform an exact binomial test to evaluate which method performs statistically significantly better across the settings. In Table 3, we summarize these results. We observe that only LW outperform the baseline across all architectures with statistically significant evidence, and outperforms the other reinitialization methods in all architectures except `resnet50`. In `resnet50`, reinitialization methods except FC perform better than the baseline but with no clear winner among them. Moreover, FC performs generally better than WELS, WELSR, and DSD. It is worth reiterating, that FC is a special case of LW that corresponds to $K = 1$ and $N > 1$.

## 3.2 Effect of Experiment Design

To determine when a particular reinitialization method outperforms others, we train a decision tree classifier on the outcomes of several experiments that vary in design by, for example, number of classes, size of the training dataset, augmentation, and dropout. Every setting contains experiment runs of each of the 5 reinitialization methods in addition to the baseline for the four architectures and 6 benchmark datasets.

We use the decision tree classifier, implemented using the Scikit-Learn package (Pedregosa et al., 2011), for interpretability. We use a minimum leaf size of 7 in the decision tree and a maximum depth of 4. Figure 2

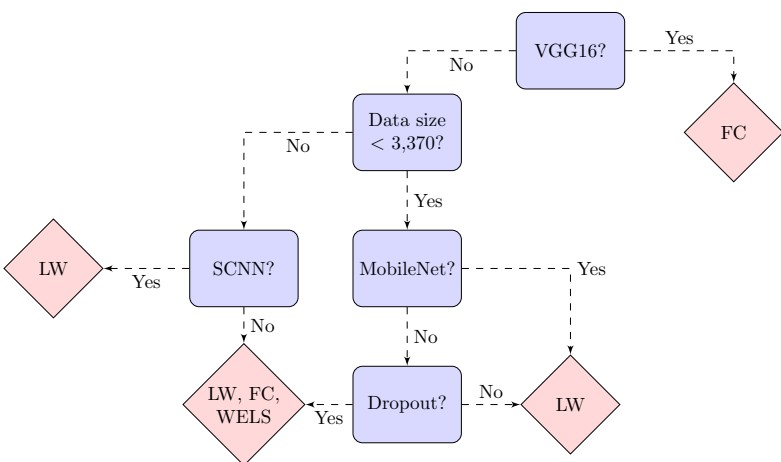

Figure 2: A decision tree classifier trained to predict the best reinitialization method based on the experiment design. The features are the training set size, number of classes, neural network architecture, dropout rate and augmentation. In general, LW performs best overall except in vgg16, where FC performs better. The decision tree classifier has a maximum depth of 4 and a minimum number sample leaf of 7.

displays the resulting decision tree. In general, LW performs best overall except in vgg16, where FC performs better.

### 3.3 Compute

In Table 2, every reinitialization round is trained until convergence. However, improvement in generalization can also be obtained at lower computational overhead by stopping early in each round. This is demonstrated in Figure 3 for all six benchmark datasets. As shown in the figure, early stopping allows to realize the gain of reinitialization without incurring significant additional overhead. In addition, we show in Appendix E that training is faster in subsequent rounds of reinitialization.

## 4 Relations to the Generalization Risk

**Boosting the Margin.** As discussed earlier in Section 2, a typical approach for bounding the generalization gap in deep neural networks is to use a particular measure of the size of the weights normalized by the *margin* on the training sample. Let $D$ be the number of layers in a neural network, whose output is a composition of functions: $f(x) = f_1 \circ f_2 \circ \cdots f_D(x)$, where each $f_i(x)$ is of the form $f_i(x) = \sigma(W_i x)$ for some matrix $W_i$ and ReLU activation $\sigma$. Then, one measure of the size of the weight that relates to generalization is the product of the Frobenius norms of layers $\prod_{i=1}^{d} ||W||_F^2$ (Neyshabur et al., 2015; 2017). This is normalized by the margin $\gamma > 0$ on the training examples, which is the smallest difference between the score assigned to the true label and the next largest score. For a better visualization, we use the margin of the softmax output in the interval $[0, 1]$.

Figure 4 displays the smallest 400 margins on the training sample for each of the benchmark datasets. As shown in the figure, LW boosts the margin on the training sample considerably when compared to previous reinitialization methods. Most importantly, LW achieves this *without* increasing the size of the weights. To take the contribution of the normalization layers into account when calculating the product $\prod_{i=1}^{d} ||W||_F^2$, we compare the product of the norms of the input to the classifier head (activations) and the norm of the weights of the classifier head in each method. We observe that LW tends to maintain the same size of the weights as the baseline. Appendix D provides further details.

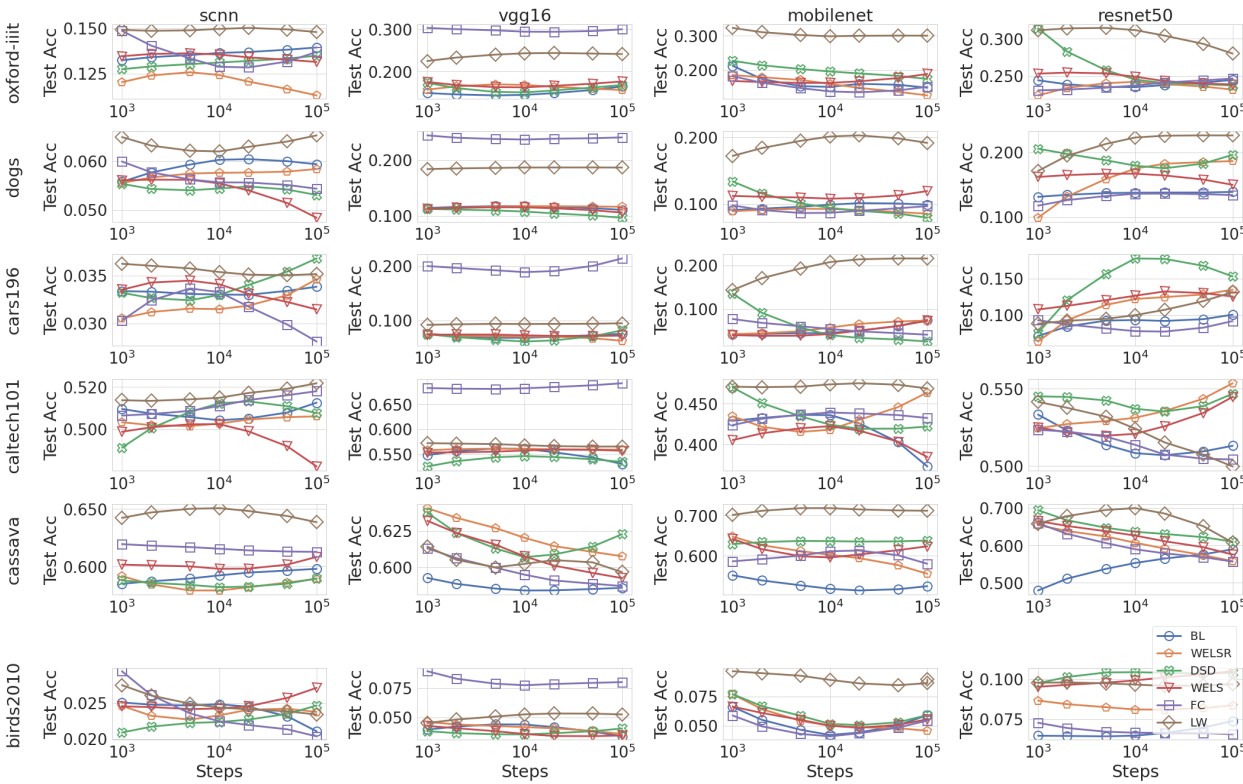

Figure 3: The test accuracy of reinitialization methods with different compute budgets (no augmentation or dropout) is plotted for each dataset. The *x*-axis is the number of training steps per reinitialization round. For the baseline, the test accuracy is plotted over the same total number of steps as reinitialization methods. Most reinitialization methods quickly surpass the accuracy of the baseline for the same amount of compute and can reap benefit of reinitialization without having to train until convergence in each round.

We provide an informal argument for why this happens in LW. First, the product of the norms of the weights in the identified $K$ blocks in LW (cf. Figure 1 and Algorithm 1) tend to remain unchanged due to the normalization layers inserted after each round. What changes is the norm of the *final* layers (following block $K$), but their norm tends to shrink because they train from scratch faster with each round. As for the margin, because the network classifies all examples correctly in a few epochs in the final round of LW, any additional epochs have the effect of increasing the margin to reduce the cross entropy loss.

**Sharpness of the Local Minima.** Finally, we observe that the final solution provided by LW seems to reside in a "flatter" local minima of the loss surface than in the baseline. One method for quantifying flatness is to compare the impact on the training loss when the model parameters are perturbed by Gaussian noise, which has been linked to generalization (Neyshabur et al., 2017). To recall, both LW and BL share the same size of the weights (cf. Appendix D). Figure 5 shows that the solution reached by LW is more robust to model perturbation than in standard training. More precisely, for every amount of noise added into the model parameters **w**, the change in the training loss in LW is smaller than in standard training suggesting that the local minimum is flatter in LW.

## 5 Discussion and Limitations

In this paper, we study a new reinitialization method for deep neural networks. Empirical results show that it improves generalization better than previous methods across a wide range of architectures and hyper-parameters. It relates to prior works that distinguish learning general rules in earlier layers from exceptions

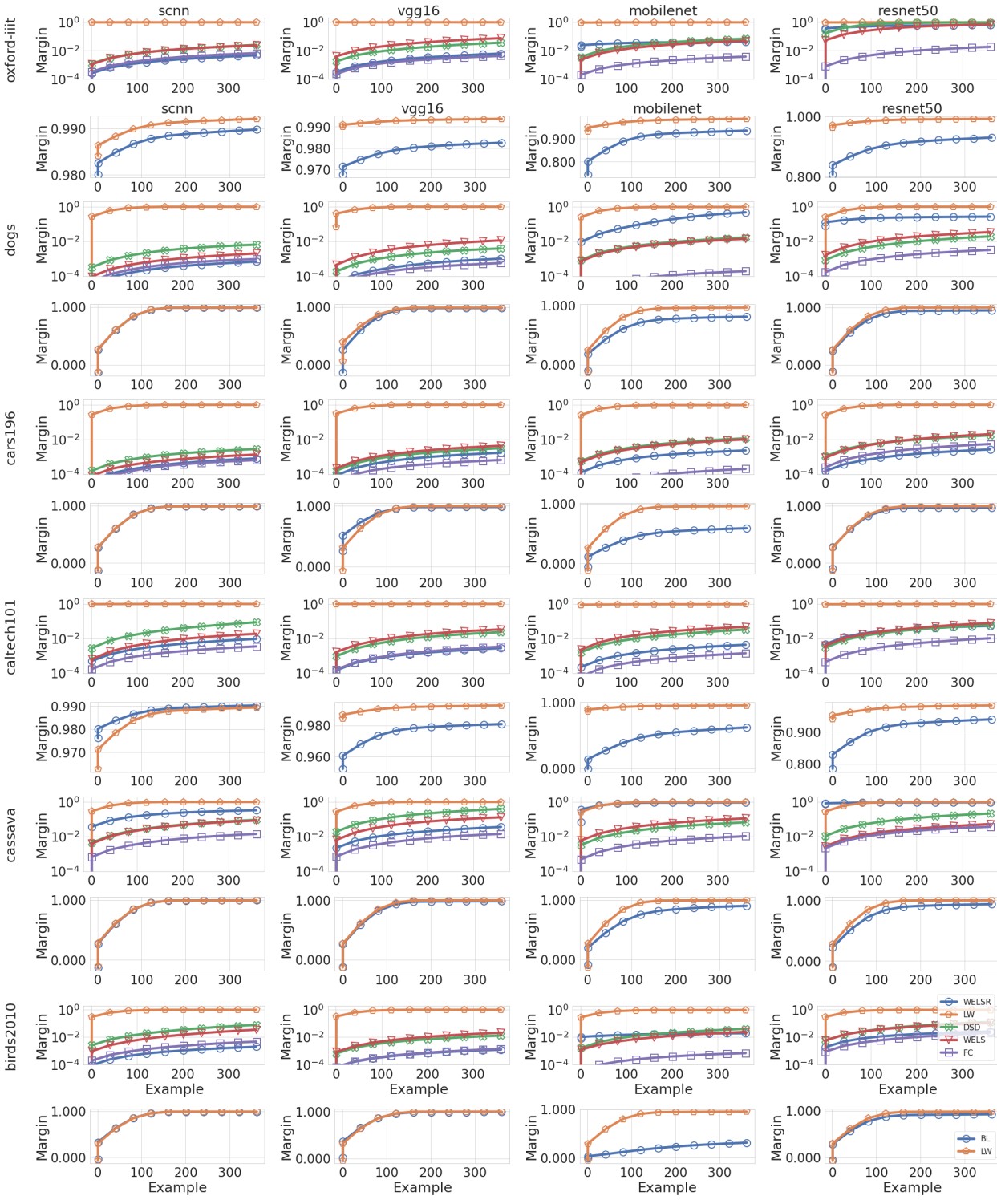

Figure 4: For each dataset, the top figure shows the smallest 400 margins in the training sample for different reinitialization methods. LW (orange) boosts the margin considerably compared to the other reinitialization methods. The bottom figure of each dataset provides the same comparison between LW and BL. The curves are displayed separately for a better visualization, as they almost coincide in the wide ranged log-scale in the top figure.

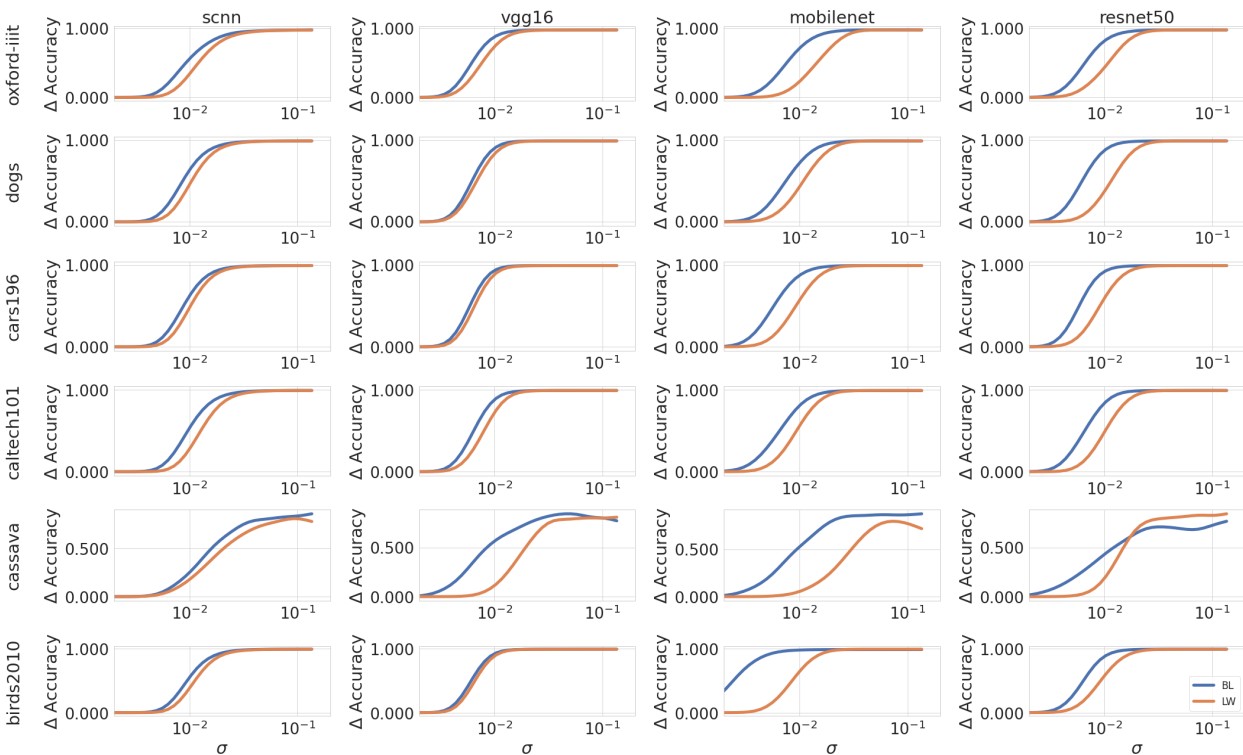

Figure 5: Bi-criteria plots for the change in training accuracy ($y$-axis) when the model parameters are perturbed by standard Gaussian noise $\mathcal{N}(0, \sigma^2 I)$ for each dataset. Lower curves suggest flatter local minima and better generalization.

to the rules in later layers, because LW places more emphasis on the early layers of the neural network. We also argue that the improved generalization can be connected to the sharpness of the local minima and the margins on the training data.

To assess the limitations of the proposed method, we conducted ablation studies, statistical tests as well as failure analysis using decision trees. Those revealed that layerwise reinitialization yields a significant improvement in cases where the generalization gap is large, such as when using poor hyper-parameters or small datasets. The improvement is small, however, when the generalization gap is small, such as when the training data is large.

Our takeaway message is that the accuracy of convolutional neural networks can be improved for small datasets using bottom-up layerwise reinitialization, where the number of reinitialized layers may vary depending on the available compute budget. At one extreme, one would benefit from reinitializing the classifier's head alone, but reinitializing all layers in sequence with rescaling and normalization yields better results. We hope that the description of the observed positive effects will inspire others to study them more and to develop more efficient alternatives.

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

# A    Layerwise Reinitialization as a Stochastic Gradient Descent

Before presenting our empirical study of the different reinitialization methods, we discuss briefly why reinitialization can be interpreted as a stochastic gradient descent (SGD) procedure to a well-defined stochastic loss. Later in Section 4, we present arguments for the improved generalization in LW that are linked to the margin on the training examples as well as the flatness of the local minimum.

Consider the following simplified training protocol. In a multi-layer neural network, let $w_0 \in \mathbb{R}^{d_0}$ be the set of weights at the first layer and write $\bar{w}_0 \in \mathbb{R}^{d-d_0}$ for the set of weights at all later layers. Let $w_0^t$ and $\bar{w}_0^t$ be the set of weights after round $t$. We will simplify discussion by denoting $w^t = (w_0^t, \bar{w}_0^t) \in \mathbb{R}^d$ and focusing on the first layer alone.

Given a loss function $\mathcal{L} : \mathbb{R}^d \to \mathbb{R}$, training via stochastic gradient descent (SGD) leads to a stationary point of the loss surface (i.e. a solution $w$ such that $\nabla \mathcal{L}(w) = 0$). Let $\mathcal{S}$ be the set of stationary points of $\mathcal{L}$. To mimic the behavior of LW, suppose that training proceeds at round $t$ by reinitializing $\bar{w}_0$ and applying the *proximal* operator to the full set of weights $w$:

$$w^t = \arg \min_{w \in \mathcal{S}} ||w - w^{t-1}||_2^2. \tag{2}$$

Informally, (2) selects a stationary point of $\mathcal{L}$ that is nearest to the current solution in the reinitialization round. If we write $\Psi_{\mathcal{S}}$ for the indicator function of the set $\mathcal{S}$:

$$\Psi_{\mathcal{S}}(w) = \begin{cases} 0, & w \in \mathcal{S} \\ \infty, & w \notin \mathcal{S}, \end{cases} \tag{3}$$

then a single training round in (2) corresponds to a gradient descent step to the *Moreau envelope* of $\Psi_{\mathcal{S}}$ (Parikh & Boyd, 2014), denoted $M_{\Psi_{\mathcal{S}}}$, which in this case is the distance to $\mathcal{S}$. That is, a training round in (2) is equivalent to the gradient update step:

$$w^{t+1} = w^t - \nabla M_{\Psi_{\mathcal{S}}}(w^t). \tag{4}$$

However, for the set of weights at the first layer $w_0$, reinitialization transforms the update rule in (4) into a *stochastic* gradient step $w_0^{t+1} = w_0^t - \nabla f_t(w_0^t)$, in which $f_t$ is a stochastic loss function whose randomness is derived from the randomness of $\bar{w}_0$ and satisfies:

$$f_t(w_0) = M_{\Psi_{\mathcal{S}}}((w_0, \bar{w}_0^t)) = \min_{(x,\bar{x}) \in \mathcal{S}} \left\{ ||x - w_0||_2^2 + ||\bar{x} - \bar{w}_0||_2^2 \right\}. \tag{5}$$

Hence, a single reinitialization round, where the weights of the first layer are retained while the rest is reinitialized, can be interpreted as a stochastic gradient descent step to the loss in (5), which penalizes weights at the first layer $w_0$ that change significantly when the rest of the network is reinitialized and retrained. Repeating this several times for the first layer is analogous to choosing a large value of $N \gg 1$ in LW. Once the first layer is trained, its output can be normalized before proceeding to the next layer, which is what LW achieves.

# B    Synthetic Data Experiment

We use the same type of data as described in Section 1, but look in more detail at the more difficult case $\alpha = 0.5$, this means the first three entries of the data encode the 8 possible labels as the 8 corners of the cube $[-0.5, 0.5]^3$, whereas the remaining entries are still sampled from the standard normal distribution. In addition, one may add a weight decay penalty to the task and examine the impact of rescaling alone. Specifically, we consider two cases:

- *Rescaling*: Instead of training once for $T$ epochs, we train 5 times for $T/5$ epochs, and in between we scale back all weights such that the norm of each layer matches the norm after initialization.

- *Reinitialization* ("LW"): In addition to rescaling, we re-initialize the layers above the first one in the first two rounds, above the second layer in the next two rounds, and only the top layer in the last round.

The results are shown in Table 5. We use the same type of data as described above, but focus now at the more difficult case of $\alpha = 0.5$.

We observe that one can get significantly better results with weight decay. Nevertheless, LW gives an additional benefit on top of the L2 regularization: The best baseline result is 0.77, but the best results with Rescaling or LW are at 0.89 / 0.86. Note that the best L2 penalty needs to be estimated (e.g. by cross validation) for each data set and training procedure. In this case, less L2 penalty is needed if we apply Rescaling / LW .

In this particular experiment, rescaling seems to already give the full effect but this is not generally the case in natural image datasets, in which the gain seems to be modest without reinitialization.

Table 5: Test accuracies (average of 100 runs)

| L2 penalty | Baseline | Rescaling | LW |
|---|---|---|---|
| 0.0 | 0.19 | 0.21 | 0.25 |
| 0.005 | 0.51 | 0.67 | 0.82 |
| 0.01 | 0.54 | **0.89** | 0.85 |
| 0.02 | 0.58 | 0.87 | **0.86** |
| 0.05 | **0.77** | 0.78 | 0.79 |
| 0.1 | 0.59 | 0.63 | 0.64 |

## C  Experiment Setup

Throughout the main text, we use four different architectures: one simple convolutional neural network, and three standard deep convolutional models.

In all architectures, we use weight decay with penalty $10^{-5}$. We also use layer normalization (Ba et al., 2016), implemented in TensorFlow (Abadi et al., 2015) using `GroupNormalization` layers with `groups=1`. Similar results are obtained when using Batch Normalization (Ioffe & Szegedy, 2015).

In all experiments, we use SGD as an optimizer with a learning rate of 0.003 and momentum 0.9. Also, we use a batch size of 256. All experiments are executed on Tensor Processing Units (TPU) for a maximum of 100,000 minibatch steps per reinitialization round. We resize images to $224 \times 224$ in all experiments.

**Simple CNN (`scnn`).** This architecture contains four convolutional blocks followed by one dense layer before the classifier head. The number of convolutional blocks $K$ used in this architecture is 4. Every convolutional block is a 2D convolutional layer, followed by layer normalization and ReLU activation. Precisely:

```
conv2d              32 filters
layer_norm; activation_relu

conv2d              32 filters
layer_norm; activation_relu
max_pooling2d

conv2d              64 filters
layer_norm; activation_relu

conv2d              64 filters
layer_norm; activation_relu
max_pooling2d

flatten
```

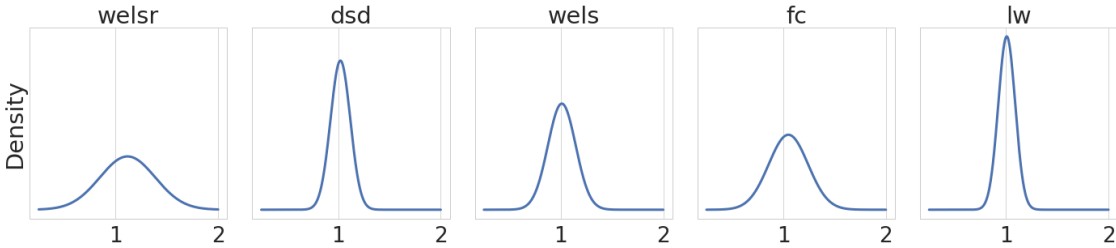

Figure 6: For each reinitialization method, the Gaussian approximation of the density of the *ratio* of the size of the weights over the size of the weights in the baseline method is shown. The density of the ratio in LW is concentrated around 1, which implies that LW tends to not increase the size of the weights. See Appendix D for details.

```
dense              512 units
layer_norm; activation_relu

dropout
classifier_head
```

**MobileNetV1 (`mobilenet`).**  This is the standard shallow MobileNet architecture (Howard et al., 2017). The standard blocks in this architecture are either convolutional blocks with layer normalization and ReLU or depthwise separable convolutions with depthwise and pointwise layers followed by layer normalization and ReLU (see Figure 3 in (Howard et al., 2017)). In the shallow architecture, the number of convolutional blocks $K$ is 7.

**VGG16 (`vgg16`).**  This is the standard VGG16 architecture (Simonyan & Zisserman, 2015). The standard blocks in this architecture are convolutional layers with layer normalization and ReLU (Table 1 in (Simonyan & Zisserman, 2015)). The number of convolutional blocks $K$ is 13.

**ResNet50 (`resnet50`).**  This is the standard ResNet50 architecture (He et al., 2016b). The standard blocks in this architecture either identity blocks or convolutional blocks (see Table 1 in (He et al., 2016b)). The number of convolutional blocks $K$ used in this architecture is 16.

## D   Size of the Weights

To calculate the norm of the weights while taking the contribution of the normalization layers into account, we compute the norm of the input to the classifier head (activations) for a random training sample of size 256. Then, we compute the Frobenius norm of the weights at the classifier head. Finally, we compute their product, which reflects the product of the Frobenius norm of layers stated in the generalization bound. Figure 6 shows a a Gaussian approximation to the ratio of the size of the weights of each reinitialization method over the size of the weights in the baseline. As shown in the figure, LW tends to maintain the size of the weights, while also boosting the margin on the training examples as discussed in the main paper.

## E   Ablation

LW includes rescaling, normalization, and reinitialization. In some cases, these may not all be required and reinitialization alone suffices, but this is not always the case. We observe a consistent improvement in LW when rescaling and normalization are included, in addition to fine-tuning the whole model at each round. In general:

- The improvement in generalization in LW cannot be attributed to rescaling or normalization alone. Reinitialization has the main effect.

- There exist experiment designs in which reinitialization fails without fine-tuning the model.

- We observe cases in which rescaling alone helps but adding reinitialization improves performance further.

- The gain from LW cannot be obtained by just training the baseline longer (i.e. using the same computational budget).

In this section, we show that the primary effect in LW comes from reinitialization, and that the improvement in generalization cannot be attributed to rescaling or normalization alone. We also show that fine-tuning the whole model performs better than freezing the early layers. Finally, we illustrate a case where LW without normalization fails.

**Rescaling.**   Generally, rescaling yields a small improvement on top of reinitialization and most of the gain of LW can often be achieved without it. Nevertheless, rescaling offers benefits. For example, in the six datasets in Table 4 plus CIFAR10 and CIFAR100, if we apply LW with rescaling vs. LW without rescaling, we observe that rescaling tends to offer better performance. The following table provides the probability (over the choice of the dataset) that rescaling yields better results compared to without it:

| | |
|---|---|
| **With no augmentation and no dropout** | 87.5% |
| **With augmentation and no dropout** | 62.5% |
| **With augmentation and dropout** | 37.5% |

In addition, one can construct settings in which reinitialization without rescaling fails. For example, when training `vgg16` on CIFAR100 without normalization layers using the following parameters:

| | | | | | |
|---|---|---|---|---|---|
| Learning Rate: | 0.003, | Momentum: | 0, | Batch size: | 256 |
| Dropout Rate: | 0, | Initializer: | He Normal, | Weight Decay: | 0, |

reinitialization fails to progress beyond random guessing without rescaling.

**Reinitialization.**   We use the `vgg16` architecture with the same hyperparameters as listed in Appendix C. We repeat the same experiments across all datasets including rescaling and normalization but without reinitialization and compare the resulting accuracy when reinitialization is added. We also include experiments with and without augmentation as well as with and without dropout. When we compare the difference in outcomes using the exact binomial test, the improvement of reinitialization compared to rescaling and normalization alone is statistically significant with a $p$-value of less than $10^{-9}$.

**Fine-tuning vs. Freezing.**   LW fine-tunes the entire model in each round. One alternative approach is to *freeze* the early blocks. However, because of the co-adaptability between neurons that arises during training (Yosinski et al., 2014), freezing some layers and fine-tuning the rest is difficult to optimize and can harm its performance (Yosinski et al., 2014). This is also true for reinitialization methods in general. Hence, the entire model including the kept layers is fine-tuned at each round. If we consider `vgg16` and the six datasets in Table 4 plus CIFAR10 and CIFAR100, for example, and apply LW with freezing vs. LW with fine-tuning, we observe that fine-tuning improves performance in general. The following table provides the probability (over the choice of the dataset) that fine-tuning the model yields better results compared to freezing:

| | |
|---|---|
| **With no augmentation and no dropout** | 87.5% |
| **With augmentation and no dropout** | 62.5% |
| **With augmentation and dropout** | 75.0% |

**Training Longer.**   The improvement in LW cannot be obtained by simply training longer even with learning rate scheduling. Throughout our experiments (e.g. Tables 2), we also train the baseline longer to have the same number of training steps in total as reinitialization methods. Despite that, reinitialization methods improve performance considerably.

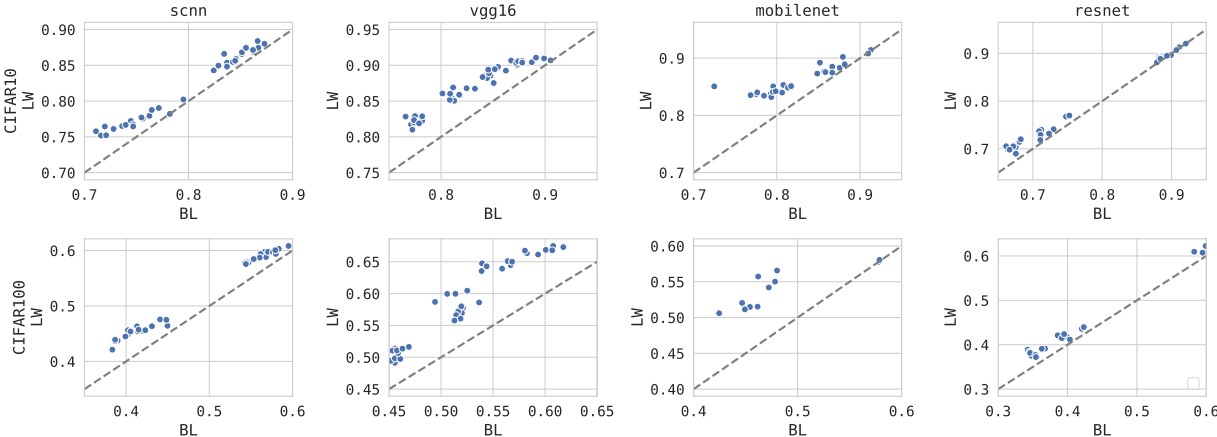

Figure 7: The test accuracy is displayed for the baseline BL ($x$-axis) vs. LW ($y$-axis) with $N = 1$ on CIFAR10 (top) and CIFAR100 (bottom). Most of the experiments fall above the diagonal, which indicates that LW succeeds in improving generalization.

## F  Experiments on CIFAR10 and CIFAR100 Datasets

To examine the impact of LW on larger datasets, we run several experiments with different hyperparamters on CIFAR10 and CIFAR100. The hyperparameters we vary are the learning rate (either 0.003 or 0.01), dropout rate (either 0 or 0.25), augmentation (with or without), and weight decay (either 1e-5 or 5e-4). The results are summarized in Figure 7. Experiments in which training fails to progress (e.g. because the learning rate is too large), were dropped. We also include experiments with Xavier initialization (Glorot & Bengio, 2010) in the figure. Since the focus in this work is on the improvement in generalization on small datasets, we only validate that LW can improve generalization compared to the baseline method BL. As shown in Figure 7, LW improves generalization in most settings, particularly when generalization is a major concern (e.g. the improvement is bigger when the baseline BL performs poorly).

