# OpenReview forum: "The Impact of Reinitialization on Generalization in Convolutional Neural Networks"
_TMLR — Rejected by TMLR_

### Review · Reviewer_rCzc · 2022-05-09

**Summary Of Contributions:**

1. The authors propose a new method to reinitialize layers of a convolutional neural networks to achieve better performance and generalization properties.

2. The proposed method (LW) involves rescaling, reinitialization and normalization, which is very simple to implement. The authors also conduct ablation studies to verify that all three components are necessary.

3. The authors study the changes of losses and margins of the trained model to verify the improvements of the generalization properties of the proposed method.


**Broader Impact Concerns:**

I don't think there is significant ethical conerns with the proposed work.

**Requested Changes:**

1. [critical] Experimental results involving larger datasets.

2. [critical] Better justifications for the rescaling component.

3. [critical] Complete ablation study results.

4. [strengthen] More theoretical analysis of the proposed method. I don't expect the paper to have a complete theoretical analysis, but it would be a strong plus.

5. [strengthen] Please provide standard deviations of the experimental results in addition to the statistical tests, as they would provide more insights into the comparisons and statistical significance.

6. [strengthen] Appendix E is not included, which should show some computation time results as intended.

7. [strengthen] Studies involve other reinitialization methods. The reinitialization component of the proposed method may use other initialization algorithms (other than He), or simply other previous reinitialization baselines such as DSD. In such a case the proposed method restricts the layers to do reinitialization.


[strengthen] Minor issues:
1. description on Layerwise (page 2): should be "At round k, the parameters at the lowest *k*" but not *K*; similarly, it should be *k* instead of *K* for the "inserted/updated following block *k*".

2. M in eq(4) is not defined.

3. This ordering of Figure 4 and Figure 5, compared to their reverse orderings in the text descriptions is a bit confusing.

4. In Appendix A, Table 5, having two red numbers for rescaling and LW while only one for baseline is a bit confusing.

**Strengths And Weaknesses:**

Strengths:
1. The proposed method is simple, easy-to-implement, and empirically effective on a range of datasets and neural network architectures.

2. The authors conduct many studies, including the synthetic data experiments, the ablation studies, the experiment design studies, and the changes in the margins/losses to further demonstrate the effectiveness of the proposed method.


Weakness:

1. The tested datasets all have relatively small sizes. I see that the proposed method probably works the best when generalization is a major concern, but I still think it's valuable to see results on some larger datasets. For example, even the overused CIFAR datasets are not included, and CIFAR datasets also exhibit some generalization issues that might get improved using the proposed method. Otherwise, the proposed method is only shown to work on small datasets, which is a major weakness.

2. I think the rescaling component is not well-justified. First of all, I don't see a clear intuition about using rescaling. Plus there are many other simple methods that can accomplish rescaling in some other way, for example, the weight decay, which is also tested in the paper. Also, the ablation studies only show the effectiveness of rescaling from one side, i.e., there are no results showing that the proposed method would work less well if rescaling is excluded.

3. The ablation studies do not have complete results. I only find qualitative statements for most of the ablation studies without quantitative numbers.

4. The theoretical analysis is relatively weak. The authors only show an interpretation of the proposed method as an SGD step, but its connections to improvements in generalization are unclear.

---

> ### Author Response · Authors · 2022-05-21
> **Response**
>
> Thank you for the detailed comments. Please see our response below.
>
> **Bigger Datasets:** We have included experiments on CIFAR10 and CIFAR100 in the revised manuscript (Appendix E). Instead of reporting the improvement of LW for a fixed set of hyper-parameters, we vary the hyperparameters here and report the improvement in each case. Please see response to Reviewer JjkS for details. In addition, small vision datasets are not uncommon in some domains, such as healthcare. For example, the state-of-the-art skin disease diagnosis tool in [1] is trained on a dataset that contains 12K examples only and the CT Image dataset in [2] contains 6K examples.
>
> **Theoretical Arguments:** As suggested by another reviewer, we have moved the SGD interpretation to the appendix to improve readability. It is quite challenging to prove theoretically that generalization improves with reinitialization in neural networks. In fact, explaining why neural networks generalize in the first place remains an open question. For this reason, we aimed to validate the improvement in generalization by conducting hundreds of experiments to verify the method and identifying explanations for the improved generalization that are supported by experiments (e.g. converging to flatter minima and boosting the margin).
>
> **Statistical Significance:** We added confidence intervals to Table 2 in the revised draft. In addition, we did include statistical significance tests with correction for multiple hypotheses testing in Table 3 in the original draft. One conclusion is that only LW performs better than the baseline across all architectures.
>
> **Compute Time:** We use the same amount of compute in all methods, including the baseline. In addition, Figure 3 shows that one can often reduce the computational overhead of LW by one order of magnitude without impacting its performance. More generally, this is a design choice since one can decrease or increase the number of rounds of reinitialization depending on the available compute budget.
>
> **Ablation:** We provide more details in the revised version including a discussion about the benefits of rescaling. We agree that the improvement of rescaling is marginal when weight decay, reinitialization and normalization are all used but one can identify settings where it continues to help and construct cases where LW without rescaling fails. We provide details in Appendix E.
>
> **Typos:** Thank you for identifying the typos. We fixed them.
>
> [1] Liu, Yuan, et al. "A deep learning system for differential diagnosis of skin diseases." Nature medicine 26.6 (2020): 900-908.
>
> [2] Cho, Junghwan, et al. "How much data is needed to train a medical image deep learning system to achieve necessary high accuracy?", ICLR, 2016.

---

### Review · Reviewer_Rx9Y · 2022-05-10

**Summary Of Contributions:**

This paper studies how the initialization of neural networks, especially CNN, will matter for generalization performances, over the empirical scope of small datasets. The analysis also leads to a new initialization strategy, namely layer-wise initialization, that can potentially leads to better results. The newly proposed method is also validated by the small scale datasets.

**Broader Impact Concerns:**

None noted.

**Requested Changes:**

- The most straightforward remedy is obviously to use larger datasets, but I imagine the authors are facing some limitations with computing resources, so I would suggest authors to find a way to show their results will be meaningful for larger datasets.

    - Please consider to at least test the new layer-wise idea in larger data sets to show that the idea indeed can be meaningful for datasets with practical interest.

    - Maybe derive some analytical bounds with dataset as a factor, and show that current results following the derived bounds, so that we can have a knowledge of the behavior of models when larger datasets are used.

**Strengths And Weaknesses:**

- Strengths

  - The coverage of the empirical study in terms of experiments and runs are fairly impressive. The results have broadly cover a wide range of studies and datasets at the small scale.
  - The layer-wise initialization is also fairly creative. Similar techniques are rarely studied to my knowledge and can potentially make a difference.

- Weakness

  - The main weakness is, probably as expected, the limited scope in terms of the data sizes, while the manuscript is studying convolutional neural network, it is quite uncommon to use datasets that are even smaller than MNIST and CIFAR10. Thus, while the study is interesting, the results can be easily challenged (or potentially altered) with bigger datasets.

---

> ### Author Response · Authors · 2022-05-21
> **Response**
>
> Thank you for the detailed comments. Please see our response below.
>
> **Bigger Datasets:** Our focus is on small datasets, where generalization can be a major concern. Nevertheless, we have included experiments on CIFAR10 and CIFAR100 in the revised manuscript (Appendix E). Instead of reporting the improvement of LW for a fixed set of hyper-parameters, we vary the hyperparameters here and report the improvement in each case. Please see response to Reviewer JjkS for details. In addition, small vision datasets are not uncommon in some domains, such as healthcare. For example, the state-of-the-art skin disease diagnosis tool in [1] is trained on a dataset that contains 12K examples only and the CT Image dataset in [2] contains 6K examples.
>
> **Theoretical Arguments:** It is quite challenging to prove theoretically that generalization improves with reinitialization in neural networks. In fact, explaining why neural networks generalize in the first place remains an open question. For this reason, we aimed to validate the improvement in generalization by conducting hundreds of experiments to verify the method and identifying explanations for the improved generalization that are supported by experiments (e.g. converging to flatter minima and boosting the margin).
>
> [1] Liu, Yuan, et al. "A deep learning system for differential diagnosis of skin diseases." Nature medicine 26.6 (2020): 900-908.
>
> [2] Cho, Junghwan, et al. "How much data is needed to train a medical image deep learning system to achieve necessary high accuracy?", ICLR, 2016.

---

### Review · Reviewer_JjkS · 2022-05-13

**Summary Of Contributions:**

This paper investigates reinitialization methods for convolutional neural network. The paper main contributions are:
  - a new reinitialization method (LW) that proposes to reinitialize the later layers in a network.
  - an empirical study on 6 small scales benchmark that demonstrates the advantage of LW for better generalization.
 - an empirical analysis of LW that investigates its effect on the network margin and flatness of the final solution.


**Broader Impact Concerns:**

No specific concern.

**Requested Changes:**

- Run LW  and the baseline on other datasets with more datapoint (CIFAR-10, CIFAR-100, Tiny-ImageNet and possibly ImageNet)
- Compare re-initialization methods with a baseline having a tuned weight-decay.
- Optimize the hyperparameters independently for a subset of the approaches (LW and the baseline).
- Add other metrics to investigate the flatness of the solution such as the spectral norm of the Hessian or Fisher Information matrix.
- While the authors discuss the computation budget, it would be nice to report the wall-clock time necessary to train the different approaches.


 Presentation could also be improved. For instance, it is not clear how section 3.1 connects with the rest of the paper. Additionally, the current version of the paper describes LW, including a pseudocode, directly in the introduction. Describing the approach in its own section could ease the reading.



**Strengths And Weaknesses:**

Strengths:
 - The paper performs an extensive set of experiments, consider various neural network backbone as well as re-initialization methods.

Weaknesses:
- Authors claim that LW improve the generalization of a convolutional neural network. However, authors focus their experimental study on small scale datasets having less than 12K training points. It is unclear if the generalization effect would be observed on larger scale dataset.
- Authors argue that LW allows to reach solution with lower-parameters norm. Weight-decay is another common method to constrain the parameters norm. While the experiments use some weight-decay, no hyperparameter search is done on its value.
- The hyperparameters are kept the same across the approaches. They might be sub-optimal for some methods.
- Authors measure the flatness at a minimum in an indirect way, by perturbing the model parameters by Gaussian noise. It would be informative to investigate other measure of the flatness a the minimum.

---

> ### Author Response · Authors · 2022-05-21
> **Response**
>
> Thank you for the detailed comments. Please see our response below.
>
> **Bigger Datasets:** We have included experiments on CIFAR10 and CIFAR100 in the revised manuscript (Appendix E). Instead of reporting the improvement of LW for a fixed set of hyper-parameters, we vary the hyperparameters here and report the improvement in each case. The hyper-parameters are chosen from the set: Augmentation (either Yes or No), Dropout (0 or 0.25), Learning Rate (either 0.003 or 0.01), Weight Decay (either 1e-5, 5e-5, 1e-4, or 5e-4), and initializer (either He normal of Xavier uniform). Experiments in which the baseline failed to reach 100% training accuracy are dropped (e.g. because the learning rate is too large). LW improves generalization in most settings. In addition, small vision datasets are not uncommon in some domains, such as healthcare. For example, the state-of-the-art skin disease diagnosis tool in [1] is trained on a dataset that contains 12K examples only and the CT Image dataset in [2] contains 6K examples.
>
> **Lower Weight Norm:** Our argument is that the ratio between the margin on the training examples over the norm of the weights is small. The reason we specifically focus on the ratio is that it has been linked to generalization using the PAC-Bayes method as we discuss in Section 4 and it relates to flatness. We do not do hyper-parameter search on the weight decay parameter because we have conducted thousands of experiments and hyper-parameter search would increase the number of experiments by an order of magnitude. However we do include it in the CIFAR experiments in Appendix E.
>
> **Flatness Measures:** The measure we use is connected to generalization in an explicit bound [3]. Please note that other notions of flatness, such as the norm of the Hessian, have been questioned in the literature since they can be changed arbitrarily without impacting the decision boundary [4].
>
> **Readability:** Thank you for pointing this out. We have moved Section 3.1 to the appendix to improve readability. We will keep the description of LW early in the paper since we refer to it in the synthetic data experiments.
>
> **Compute Time:** We use the same amount of compute in all methods, including the baseline. In addition, Figure 3 shows that one can often reduce the computational overhead of LW by one order of magnitude without impacting its performance. More generally, this is a design choice since one can decrease or increase the number of rounds of reinitialization depending on the available compute budget.
>
> [1] Liu, Yuan, et al. "A deep learning system for differential diagnosis of skin diseases." Nature medicine 26.6 (2020): 900-908.
>
> [2] Cho, Junghwan, et al. "How much data is needed to train a medical image deep learning system to achieve necessary high accuracy?", ICLR, 2016.
>
> [3] Neyshabur, Behnam, et al. "Exploring generalization in deep learning." NeurIPS, 2017.
>
> [4] Dinh, Laurent, et al. "Sharp minima can generalize for deep nets."ICML, 2017.

---

### Decision · Action_Editors · 2022-06-19

**Recommendation:** Reject

**Comment:**

The manuscript was evaluated by three reviewers. They found that the manuscript has some strengths, particularly the creativity of the proposed approach and the scope of experiments on small datasets, but that these strengths were surpassed by several weaknesses, particularly the small scale of the considered datasets, insufficient justification of the proposed approach, insufficient analysis and discussion of parameter norms, flatness of minima, hyper parameters, and the limited theoretical results. The authors' responses addressed some of the initial concerns, particularly by adding new experiments and reorganising parts of the manuscript to improve readability. However, the reviewers found that important recommendations remained unaddressed and important issues remained unresolved, particularly about the justification of the rescaling approach and the limited theoretical results. At the end of the discussion period there was a consensus that the article requires more results to justify the approach, with final recommendations ranging from leaning reject to reject. Hence I must reject the article in its current form. However, I would be willing to consider a significantly revised version addressing the feedback from the reviewers.